# Visual Category Discovery via Linguistic Anchoring

## Abstract

We address the problem of generalized category discovery (GCD) that aims to classify entire images of a partially labeled image collection with the total number of target classes being unknown. Motivated by the relevance of visual category to linguistic semantics, we propose language-anchored contrastive learning for GCD. Assuming consistent relations between images and their corresponding texts in an image-text joint embedding space, our method incorporates image-text consistency constraints into contrastive learning. To perform this process without manual image-text annotations, we assign each image with a corresponding text embedding by retrieving $k$-nearest-neighbor words among a random corpus of diverse words and aggregating them through cross-attention. The experimental evaluation shows the promising performance of the proposed method on standard benchmarks, ImageNet100, CUB, Stanford Cars, and Herbarium19.

## 1 Introduction

Clustering is one of the most fundamental problems in unsupervised learning, which aims to assign similar instances from a data collection to the same cluster while separating distinctive instances to different clusters. The problem of clustering is similar to drawing decision boundaries on a feature space and thus naturally links to classification, but typically does not assume specific target classes. The spirit of clustering is basically unsupervised, however, in a practical scenario, a subset of the target classes can be given as supervision such that a model learns to cluster data points and discover the remaining classes. Based on this concept, recent work (Vaze et al., 2022) introduces such an experimental setup of generalized category discovery (GCD). In other words, GCD is essentially a clustering problem with a few additional supervision, which motivates us to introduce an effective and general clustering algorithm that tackles image clustering and GCD at the same time.

We propose language-anchored contrastive learning for GCD that leverages vision and language (VL) consensus. Our approach is motivated by the fact that word-level texts may play the role of a robust anchor for image clustering; for example, we can imagine a huge number of different cat visuals with pixel-level differences, whereas describing a cat in a single word would obviously have significantly fewer degrees of freedom. Inspired by this, we introduce an image clustering objective based on VL consensus in a label-efficient manner – using neither ground-truth class text labels nor a collection of image-text pairs. Our method first builds a text memory with an external source of arbitrary word-level texts. We then establish unsupervised VL correspondence of an image with $k$ nearest-neighbor ($k$NN) search on the text memory in order to incorporate the relevant words for better clustering. The relevant words are aggregated based on the query-text similarity and thus considered a set of basis for representing the query image. Using this aggregated embedding as a linguistic anchor, we design a contrastive loss based on the hypothesis of VL consensus; relations between images and their corresponding texts should be consistent in an image-text joint embedding space. The contrastively trained image embeddings are clustered at the end for classification. While the prior work Vaze et al. (2022), Zhang et al. (2023), Pu et al. (2023) uses $K$-means clustering, we observe that the use of ground-truth number of classes is critical to the previous methods during model selection. In contrast, leveraging the VL consensus, our method outperforms them on the public benchmarks of GCD even without using the number of target clusters.

The contribution of our work can be summarized as follows:

- We propose *language-anchored contrastive learning* that clusters images with the guide of their corresponding text anchor in the absence of the annotated image-text grounded pairs.

- Our model performs clustering with the total number of target classes being completely unknown, which has been overlooked by previous work on GCD.

- The proposed model achieves the state-of-the-art performance on the public GCD benchmark of coarse-graind and fine-grained datasets.

## 2 RELATED WORK

### 2.1 GENERALIZED CATEGORY DISCOVERY

GCD, formulated by Vaze et al. (2022), aims to classify images when only a subset of known classes are labeled, without knowing the number of the target classes a prior. It is heavily motivated by NCD (Han et al., 2019) but relaxes its assumption that unlabeled images only consist of unknown classes with the given number of the target classes. The underlying concept of this task is to enhance image clustering by transferring the knowledge learned from labeled images to unlabeled images. In this sense, GCD (Vaze et al., 2022) suggests a semi-supervised contrastive learning framework, while other methods further explore various pseudo-labeling mechanisms for unlabeled images. For example, DCCL (Pu et al., 2023) adopts InfoMap clustering to leverage and update pseudo-assignment of images during training, while PromptCAL (Zhang et al., 2023) discovers pseudo-positive samples based on semi-supervised affinity generation.

Although these methods show remarkable progress by enhancing the framework of contrastive learning with partially labeled images, we notice that most have neglected the unknown number of the target classes in implementation. Such methods assume known classes during model selection and hyperparameter search, which is an unrealistic scenario that conflicts with the fundamentals of the task. Furthermore, jointly estimating the number of classes has been less explored so far, leaving this stage as a separate branch from training, while a concurrent work (Zhao et al., 2023) tackles this in a parametric way. Pu et al. (2023) is the first to propose the joint framework, while it requires heavy computation by clustering the entire train set, and the clustering threshold is manually chosen by referring to both known and unknown classes.

### 2.2 KNOWLEDGE GUIDED VISION AND LANGUAGE MODELS

Recently, image-text foundation models such as CLIP have been thoroughly exploited for various image recognition tasks in direction of engaging external text information with images. To leverage the clear alignment between two modalities, large-scale text memory or well-refined image-text pairs are taken for granted. For example, RA-CLIP (Xie et al., 2023), LGSimCLR (El Banani et al., 2023) and RECO (Iscen et al., 2023) assume a large pre-defined image-text pairs to retrieve relevant images for contrastive learning. In contrast, Wei et al. (2023), Liu et al. (2023) consider unpaired image and text information, yet still require specific knowledge in advance, such as the name of target classes (Wei et al., 2023) or the knowledge about the downstream dataset (Liu et al., 2023). Unlike these work, our method does not require any image-text annotations, nor the knowledge about the dataset we aim to classify. We also assume a word corpus obtained by a caption dataset, which is more lightweight than large-scale or web-scale memory.

## 3 OUR APPROACH

Our motivation is that visual categories correspond to linguistic semantics in principle. We thus leverage two modalities of vision and language using the pre-trained CLIP encoder (Radford et al., 2021) as a base feature extractor for image and text, and introduce language into image clustering but in the absence of the annotated pairs of images and texts. Using the query image feature, we retrieve its $k$ nearest texts from a text feature corpus of arbitrary-collected natural language words (section 3.1). The retrieved word set, which is unordered and noisy, is then aggregated to a single text feature, which we call text prototype, through attentive pooling with respect to the query similarity. To leverage the consensus between modalities as well as within a modality, we formulate

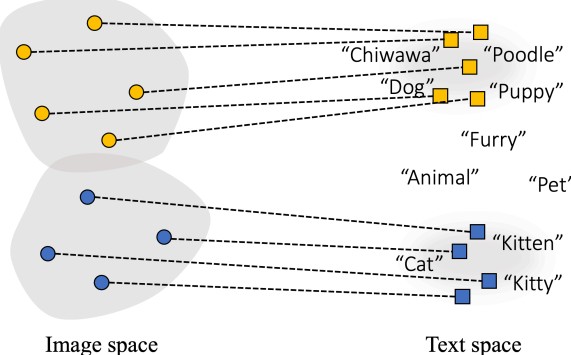

Figure 1: **Vision and language cluster consensus.** Our hypothesis suggests that the visual variances in the image space can be concentrated in the text space by leveraging the relevant semantic concepts as an anchor.

a constrastive learning loss that incorporates the cross-modal relation, which we call *Vision-and-Language Contrastive Loss*, or VLC loss(section 3.2). Moreover, we propose to jointly estimate the unknown number of classes via agglomerative clustering which allows us to evaluate our method in more practical scenario(section 3.3).

**Problem definition.** GCD aims to classify images when a partially labeled images are given without knowing the number of target classes in advance. In other words, the dataset consists of labeled-known images, unlabeled-known images, and unlabeled-unknown images. Formally, a train set $\mathcal{D}_T$ consists of a labeled set $\mathcal{D}_\mathcal{L} = \{(x_i, y_i)\}_{i=1}^N \in \mathcal{X} \times \mathcal{Y}_\mathcal{L}$ and an unlabeled set $\mathcal{D}_\mathcal{U} = \{(x_i, y_i)\}_{i=1}^M \in \mathcal{X} \times \mathcal{Y}_\mathcal{U}$, where $\mathcal{Y}_\mathcal{L} \subset \mathcal{Y}_\mathcal{U}$. Note that the number of target classes $|\mathcal{Y}_\mathcal{U}|$ is unknown. A validatation set $\mathcal{D}_V$ follows identical structure, whereas the final performance is tested by clustering a train set $\mathcal{D}_T$, and then evaluating the accuracy score on $\mathcal{D}_\mathcal{U}$.

### 3.1 RELEVANCE OF VISUAL CATEGORY TO LINGUISTIC SEMANTICS

In this section, we describe the motivating observation to leverage the consensus of the two modalities based on the pretrained VL embedding space of CLIP. We justify the underlying assumption about the relation between a given image and the relevant text information collected by $k$ nearest-neighbor ($k$NN) search, showing that the related word set plays an important role as a semantic anchor for image clustering.

Considering that the text memory we assume consists of word-level semantics, we hypothesize that the semantic similarity between two images is better captured by corresponding texts in the linguistic embeddings. In other words, the visual variances expressed in the image space can be grounded by the corresponding language concepts, such as its name. Such a property is well known so far, yet we examine the hypothesis in a more generalized setup, using an image feature and the average of $k$-nearest neighbor words instead of ground-truth annotation. Figure 1 visualizes such an example.

To quantify the property, we compare the ratio of *inter*-class variance to *intra*-class variance for each modality; a higher value implies a better distribution for clustering. The image space achieves 0.92 while the text space achieves 1.10. We also examine the silhouette score, which diagnoses how well the data points are separated from the nearest clusters compared to the cluster to which they belong. The silhouette scores are 0.09 and 0.12 for image and text, respectively. Interestingly, the language modality is superior on both the variance ratio score and the silhouette score, indicating that our hypothesis holds even without using well-defined image-text pairs.

Based on the results, we establish image-text pairs for our approach by retrieving the $k$-nearest neighbors ($k$NN) from an arbitrary word corpus. Note that the established image-text pair $(i, \mathbb{T})$ is not static over training, instead, we learn to pair them better by training the last layer of the vision encoder implicitly through the end-task learning objective. In this manner, the image-to-text-set correspondence is established without image-text grounded annotation cost, which is equipped with a pair of arbitrary collections of images and word texts.

Figure 2: **Overview of our method.** For a given image feature, we retrieve its $k$ nearest texts from a text feature corpus. The retrieved texts are aggregated by cross-attention mechanism, and the image features are fed to self-attention mechanism as well. The generated image and text prototypes are learned to encourage its consensus in both *intra-* and *inter*-modality.

## 3.2 LANGUAGE-ANCHORED CONTRASTIVE LEARNING

In this section we introduce a framework to associate vision and language consistency in a learnable manner. We propose two branches of attentive prototype learning for each modality by adopting attention mechanism, and suggest *Vision-and-Language Contrastive Loss* which jointly incorporates image and text prototypes.

**Attentive learning of text sets.** As formulated in the previous paragraph, our method leverages set-based similarity, and here we present a learnable architecture to represent a set. Recall the text elements are word-level units from an arbitrary word corpus, hence, the retrieved $k$NNs often poorly or too broadly ground the image. Given such a coarsely-grained set of $k$NN words, we present a learnable transform of the text set $\mathbb{T}$ into an embedding $t$ such that it adaptively reflects the semantic aspects of the input image the most. We choose to use the cross-attention mechanism based on image-text similarity, where the query and key input to the attention layer corresponds to the input image feature $h$ and its $k$NN text set $\mathbb{T}$, respectively, sharing the value input with that of the key. The output embedding $t$, which we call *text prototype* hereafter, then reflects the cross-modal correlation between the image in aggregating the set of text embedding. In addition, the noisy elements in $k$NN are effectively filtered out through the learning procedure.

Through the image-text paring, an image is associated with a text set. We also consider amplifying more plausible text pairs for an image by *text swap augmentation*. In detail, we randomly switch the text set of a given image with the set from the augmented image, and empirically found that random switching broadens the consensus beyond exploiting naive nearest neighbors.

**Attentive learning of image instances.** Recall that our pipeline follows the text $k$NN retrieval, which is explained in an earlier paragraph, and then image feature learning with the incorporation of the retrieved texts. In other words, an input image feature is used with two different objectives of cross-modal (instance-to-instance) text retrieval and cluster feature learning with the retrieved text set retrieval (instance-to-set). To this end, we add another attention layer that transforms the image embedding $h$ from the CLIP backbone, which we use for the text retrieval, and produces another level of representation $v$ of the same dimension for clustering. Note that the $v$ will directly be aligned with text prototypes and eventually used for clustering, which will be described in the following paragraph.

**Vision-and-language contrastive loss.** For learning objective, we leverage the framework of contrastive learning in terms of both inter-modality and intra-modality. In detail, we consider two views of the given image: original image $i$ and its randomly augmented image $i'$ with crop, flip and color jittering , obtaining image prototypes $v_i, v'_i$ and text prototypes $t_i, t'_i$ for each view. Next, for the prototypes of labeled images, we formulate supervised contrastive loss:

$$\mathcal{L}_v^i = -\frac{1}{|\mathcal{N}(i)|} \sum_{p \in \mathcal{N}(i)} \log \frac{\exp(v_i \cdot v_p / \tau)}{\sum_{j \notin \mathcal{N}(i)}^{B} \exp(v_i \cdot v_j / \tau)}, \tag{1}$$

where $\mathcal{N}(i)$ denotes samples in a batch $B$ which belong to the same classes of an image $i$, and $\tau$ as a temperature value. Through this, a learner can gain clear knowledge within visual modality, which ensures more stable learning. The clear alignment learned from labeled images is then transferred to entire images through cross-fusion contrastive loss. Cross-fusion contrastive loss regards fused

embeddings $c_i$ and $c_i'$ as a positive pair as well, where $c_i$ is obtained by a sum of image prototype $v_i$ and text prototype $t_i$:

$$\mathcal{L}_{cross}^i = -\log \frac{\exp((v_i \cdot v_{i'} + t_i \cdot t_{i'} + c_i \cdot c_{i'})/\tau)}{\sum_{j \neq i}^B \exp((v_i \cdot v_j + t_i \cdot t_j + c_i \cdot c_j))/\tau)}, \tag{2}$$

$$\text{where, } c_i = v_i + t_i.$$

This can be also interpreted as engaging both *inter*-modality and *intra*-modality terms in a combined form Equation 2, inducing consistent vision and language alignment. Furthermore, combining these terms into a single equation rather than using multiple separate losses maintains more active interaction among them, since any misalignment among these terms penalizes the other terms as well, with bigger weights due to the shape of exponential function.

The overall loss is then formulated as:

$$\mathcal{L}_{VLC} = \lambda \mathcal{L}_v + (1 - \lambda)\mathcal{L}_{cross}. \tag{3}$$

where $\lambda$ denotes the weights of supervised contrastive loss.

### 3.3 CLUSTERING WITH CLUSTER NUMBER ESTIMATION

We revisit agglomerative clustering to integrate two separate branches of fine-tuning image representations and estimating the unknown number of classes K. As the previous work empirically showed, the motivation is that given a set of embeddings with unknown K, clustering accuracy is proportional to how close the estimated K is to the ground-truth. Starting from this observation, at the end of every epoch, we compare the clustering accuracies of labeled images by differing K, and choose the one with the highest accuracy score. Moreover, the highest accuracy score corresponds of the chosen K can be used as a criterion for validation, unlike previous work which assumes a given number of K in this stage. Note that only a subset of known classes are labeled during validation stage as same as training stage.

The remaining question is how can it be applied in a single pipeline in an effective way. A naive approach would be to iteratively run conventional clustering algorithms (e.g., k-means), which is hardly applicable during training stage in terms of computational efficiency. Instead, we suggest taking advantage of a linkage matrix of given embeddings used for agglomerative clustering. Once we get a linkage matrix referring to the distance of the nearest clusters(or samples), we can incrementally split the clusters by differing the distance threshold, without requiring any additional clustering operations.

## 4 EXPERIMENTS

In this section we evaluate our method on GCD and carefully analyze the contribution of each framework. For the main experiments, we compare the clustering accuracy on GCD and inductive GCD setup with various benchmarks. We also report the number of estimated number of classes K and compare with other method.

### 4.1 EXPERIMENTAL SETUP

**Datasets.** We evaluate our method on 6 image classification benchmarks; 2 generic datasets CI-FAR100 (Krizhevsky & Hinton, 2009), ImageNet-100 (Geirhos et al., 2019) and 4 fine-grained datasets CUB-200 (Wah et al., 2011), Stanford Cars (Krause et al., 2013), Aircraft (Maji et al., 2013), and Herbarium19 (Tan et al., 2019). For splitting the target classes to known and unknown set, we adopt the SSB (Vaze et al., 2021) split for CUB-200, Stanford Cars, and Aircraft benchmarks, and random sampled with the same seed of GCD (Vaze et al., 2022) for the other benchmarks. To sample a subset of labeled images from the known classes, we follow Vaze et al. (2022) and set the ratio as 80% for CIFAR-100 and 50% for other datasets. For a word memory, we adopt Google Conceptual Captions 12M dataset (Sharma et al., 2018) and split it into a set of words. One thing to note is that the target classnames are contained in the text set sometimes, but we do not filter them out since excluding unknown target classes beforehand is unrealistic.

**Training details.**   We use pretrained CLIP ViT-B/16 as a backbone, and train the last layer of the image encoder. All experiments are done on 2 RTX-3090 GPUs with a batch size of 256. For constructing $k$-NN text sets, the cardinality of $k$ is set to 10, and updated every epoch. The weight of supervised contrastive loss $\lambda$ is set to 0.2 for generic classification datasets(CIFAR100 and ImageNet-100), and 0.35 for fine-grained classification datasets. All the other hyperparameters including learning rate, weight decay, and the number of augmented images are adopted same as GCD (Vaze et al., 2022).

**Baselines.**   We compare our method with 3 previous work, GCD (Vaze et al., 2022), Prompt-CAL (Zhang et al., 2023) and DCCL (Pu et al., 2023). For a fair comparison, we retrain GCD and PromptCAL by switching the backbone from DINO (Caron et al., 2021) ViT-B/16 to pretrained CLIP ViT-B/16 image encoder. Note that the backbone adopted by the previous work is pretrained on ImageNet1K, which conflicts with one of the benchmark ImageNet-100. For reproducing Prompt-CAL with the switched backbone, a learning rate is set to 0.01 as the performance is largely degraded when using the original value.

**Evaluation metrics.**   We evaluate the performance on K-means clustering with the image prototypes generated by the self-attention module. The accuracy score is computed on the clusters assigned with ground-trugh classes by Hungarian algorithm, followed by GCD (Vaze et al., 2022). We report the accuracy on both ground-truth number K and the estimated K, as the previous work assumes the given number of target classes during evaluation.

## 4.2   MAIN RESULTS

**Evaluation on GCD.**   Table 1 compares the results on GCD setup in both generic and fine-grained classification benchmarks. In GCD, a test set is used for validation and the performance is evaluated on an unlabelled train set. Note that unlike other baselines, we assume an unknown number of classes during validation.

On the generic classification benchmarks, our method shows great improvements especially in ImageNet100. Specifically, the accuracy is comparable on CIFAR100 and 4.9% higher on ImageNet-100 with the given number of target classes. The performance is consistent even when the estimated number of classes is used for clustering as well. We also investigate the effectiveness on the fine-grained classification benchmarks, which shows state-of-the-art performance on most cases. Especially, our method achieves 3.4% higher, 3.9% higher on CUB and Stanford Cars respectively. From this, we validate that incorporating text knowledge benefits on image clustering by ensuring the consistency between relevant images.

**Evaluation on inductive GCD.**   We also compare the clustering results on inductive setup followed by Zhang et al. (2023). Compared to GCD task which focuses on a transductive scenario, we use a part of a train set for validation and evaluate our model on a test set in this task. As shown in Table 2, our method outperforms on both generic and fine-grained datasets overall. The results imply that our method can be generalized to unseen images during training as well, unlike conventional image clustering methods.

**K Estimation.**   We also compare the estimated number of classes K in Figure 3 with the ground-truth number and the result reported by DCCL, which jointly estimates the number of classes during the training stage as well. The reported number is averaged among 3 trials for each benchmark. Noticeably, our method shows better accuracy overall without requiring any hyperparameters dependent to the target benchmark. Note that the value is approximated on a test set, which is computationally efficient compared to DCCL which refers to the entire training set.

## 4.3   DESIGN CHOICE ANALYSIS

In this section, we examine the contribution of each module and learning objective through extensive experiments. Note that all experiments are conducted on ImageNet100 benchmark if not specified.

**Effectiveness of cross-attention.**   To transform a set of retrieved words, we adopt cross-attention mechanism which incorporates visual information with word-level semantic concepts. Instead of

Table 1: Clustering results for GCD

| Method | Backbone | CIFAR100 | | | ImageNet100 | | | CUB | | |
|---|---|---|---|---|---|---|---|---|---|---|
| | | All | Old | Novel | All | Old | Novel | All | Old | Novel |
| GCD | DINO ViT-B/16 | 73.0 | 76.2 | 66.5 | 74.1 | 89.8 | 66.3 | 51.3 | 56.6 | 48.7 |
| GCD | CLIP ViT-B/16 | 70.4 | 79.3 | 52.6 | 71.6 | 86.0 | 64.6 | 51.1 | 56.2 | 48.6 |
| DCCL | DINO ViT-B/16 | 75.3 | 76.8 | 70.2 | 80.5 | 90.5 | 76.2 | 63.5 | 60.8 | 64.9 |
| PromptCAL | DINO ViT-B/16 | **81.2** | 84.2 | **75.3** | 83.1 | 92.7 | 78.3 | 62.9 | 64.4 | 62.1 |
| PromptCAL | CLIP ViT-B/16 | 69.4 | 77.3 | 53.5 | 75.2 | 87.0 | 69.3 | 53.7 | 61.4 | 49.9 |
| Ours (estimated $K$) | CLIP ViT-B/16 | 80.2 | 82.6 | **75.3** | **88.5** | **95.9** | **84.7** | 57.3 | 49.4 | 61.2 |
| Ours | CLIP ViT-B/16 | 80.8 | **84.6** | 73.1 | 88.0 | **95.9** | 83.9 | **66.9** | **69.0** | **65.8** |

| Method | Backbone | Stanford Cars | | | FGVC Aircraft | | | Herbarium 19 | | |
|---|---|---|---|---|---|---|---|---|---|---|
| | | All | Old | Novel | All | Old | Novel | All | Old | Novel |
| GCD | DINO ViT-B/16 | 39.0 | 57.6 | 29.9 | 45.0 | 41.1 | 46.9 | 35.4 | 51.0 | 27.0 |
| GCD | CLIP ViT-B/16 | 62.5 | 73.9 | 57.0 | 41.2 | 43.0 | 40.2 | 39.7 | 58.0 | 29.9 |
| DCCL | DINO ViT-B/16 | 43.1 | 55.7 | 36.2 | - | - | - | - | - | - |
| PromptCAL | DINO ViT-B/16 | 50.2 | 70.1 | 40.6 | **52.2** | **52.2** | **52.3** | 37.0 | 52.0 | 28.9 |
| PromptCAL | CLIP ViT-B/16 | 60.1 | **77.9** | 51.5 | 42.2 | 48.4 | 39.0 | 37.4 | 50.6 | **30.3** |
| Ours (estimated $K$) | CLIP ViT-B/16 | 64.1 | 76.2 | 58.3 | 47.6 | 42.4 | 50.2 | **40.8** | **62.7** | 29.0 |
| Ours | CLIP ViT-B/16 | **66.4** | 76.6 | **61.5** | 48.5 | 48.7 | 48.3 | 40.4 | 62.0 | 28.8 |

Table 2: Clustering results for inductive GCD

| Method | Backbone | CIFAR100 | | | ImageNet100 | | | CUB | | |
|---|---|---|---|---|---|---|---|---|---|---|
| | | All | Old | Novel | All | Old | Novel | All | Old | Novel |
| GCD | DINO ViT-B/16 | 70.1 | 76.8 | 43.5 | 79.7 | 92.7 | 66.7 | 57.5 | 64.5 | 50.6 |
| PromptCAL | DINO ViT-B/16 | **81.6** | **85.3** | 66.9 | 84.8 | 94.4 | 75.2 | **62.4** | 68.1 | **56.8** |
| PromptCAL | CLIP ViT-B/16 | 79.9 | 82.7 | 68.5 | - | - | - | 56.0 | 67.7 | 44.5 |
| Ours (estimated $K$) | CLIP ViT-B/16 | 80.1 | 83.0 | 68.9 | 84.0 | 94.7 | 73.2 | 57.9 | 68.6 | 47.4 |
| Ours | CLIP ViT-B/16 | 80.3 | 83.0 | **69.5** | **89.3** | **95.5** | **83.1** | 59.7 | **71.6** | 46.7 |

| Method | Backbone | Stanford Cars | | | FGVC Aircraft | | | Herbarium 19 | | |
|---|---|---|---|---|---|---|---|---|---|---|
| | | All | Old | Novel | All | Old | Novel | All | Old | Novel |
| PromptCAL | CLIP ViT-B/16 | 62.3 | 76.9 | **48.2** | 43.6 | 49.5 | **37.7** | 37.6 | 50.3 | 30.7 |
| Ours (estimated $K$) | CLIP ViT-B/16 | 62.5 | **83.1** | 42.7 | 44.1 | 54.0 | 34.2 | **48.7** | 57.7 | 33.1 |
| Ours | CLIP ViT-B/16 | **62.7** | 80.1 | 45.8 | **45.6** | **55.0** | 36.2 | 48.0 | **58.5** | **36.8** |

this, we explore several choices of set-to-instance transformation methods; (a) simple average and (b) simple average followed by a projection head. For (b), we adopt the projection head implemented on DINO (Caron et al., 2021), which is used in other baselines as well. The projection head consists of 3 linear layers with a GELU (Hendrycks & Gimpel, 2016) activation layer, and the hidden dimension is set to 2048. The results in Table 4 indicate that simply averaging the text embeddings does not reflect any consensus between image and text pairs, leading a significant performance drop. In the case of (b), which refers to the third line in Table 4, adopting cross-attention performs better although the projection head exploits more parameters than cross-attention. This demonstrates that the performance gain is due to associating visual information to a coarse-grained text set, not about fine-tuning the additional parameters.

Table 3: Ablation study on effectiveness of Vision-and-Language Consistency Loss.

| | $\mathcal{L}_v$ | $\mathcal{L}_{cross}$ | combined form | All | Old | Novel |
|---|---|---|---|---|---|---|
| (1) | ✓ | $\mathcal{L}_{InfoNCE}$ | | 76.1 | 93.7 | 67.2 |
| (2) | ✓ | $\mathcal{L}_{CLIP}$ | | 56.7 | 81.0 | 44.4 |
| (3) | ✓ | ✓ | | 52.7 | 78.6 | 39.6 |
| (4) | | ✓ | ✓ | 76.8 | 72.1 | 79.1 |
| (5) | ✓ | ✓ | ✓ | 88.0 | 95.9 | 83.9 |

Table 4: Ablation study on the effectiveness of the two attention mechanisms.

| | self-attn | cross-attn | All | Old | Novel |
|---|---|---|---|---|---|
| (1) | | mean | 74.0 | 88.0 | 66.9 |
| (2) | ✓ | mean | 81.2 | 92.4 | 75.6 |
| (3) | ✓ | mean proj | 84.3 | 95.9 | 78.5 |
| (4) | | ✓ | 83.1 | 96.2 | 76.6 |
| (5) | proj | ✓ | 80.1 | 96.1 | 72.1 |
| (6) | ✓ | ✓ | 88.0 | 95.9 | 83.9 |

Table 5: Ablation study on robustness against text memory.

| | | All | Old | Novel |
|---|---|---|---|---|
| (1) | w classnames | 88.0 | 95.9 | 83.9 |
| (2) | w/o classnames | 88.3 | 95.8 | 84.5 |

Table 6: Ablation study on effectiveness of text swap augmentation.

| text swap augmentation | | ImageNet100 | | | Stanford Cars | | |
|---|---|---|---|---|---|---|---|
| | | All | Old | Novel | All | Old | Novel |
| (1) | | 87.9 | 95.8 | 83.9 | 60.4 | 76.8 | 52.4 |
| (2) | ✓ | 88.0 | 95.9 | 83.9 | 66.4 | 76.6 | 61.5 |

**Effectiveness of image self-attention.** To validate the necessity of the image projection module, we compare our method by (a) learning without self-attention, and (b) replacing it to projection layers. Note that unlike self-attention, the projection head takes $[CLS]$ embedding of an given image as an input. The projection head is adopted same as the previous experiment. As shown in Table 4, learning without self-attention degrades the performance, since it no longer ensures a symmetric architecture for feature-level alignment and prototype-level alignment. Applying projection layers shows worse performance, which implies that the role of the image projection branch is not about learning fine-tuned embedding, but to be well-aligned with the corresponding text prototype. Through this, we validate that image self-attention assists the prototype-level alignment, preserving the consistency between the two modalities.

**Effectiveness of Vision and Language Consistency Loss.** Table 3 shows the ablation study for each component of the learning objective on ImageNet100. Our cross-fusion contrastive loss $\mathcal{L}_{cross}$ is compared with (a) InfoNCE loss (Oord et al., 2018) and (b) CLIP loss, which can be considered as adopting either *intra*-modality or *inter*-modality term only. Compared to InfoNCE loss, we validate that using text information as an additional positive term supports contrastive learning. Interestingly, we noticed that our method does not fit with CLIP loss, indicating that both *intra*-modality and *inter*-modality are essential when well-refined image-text annotations are unavailable. Furthermore, we analyze the effect of combining both *intra*-modality and *inter*-modality terms in a single exponential term, instead of adopting multiple separate terms. We find that combining these terms plays an important role by enforcing the alignment of the two modalities stronger. Lastly, the performance decreases without supervised contrastive loss $\mathcal{L}_v$ as well, indicating that the clear alignment learned from labeled images is essential to guide the vision-and-language consistency of unlabeled images.

**Effectiveness of text swap augmentation.** We evaluate the effect of a text set augmentation in Table 6. The performance gain is clearer on fine-grained datasets, when the retrieved text sets are likely to be noisy and insufficient. From this result, we can clarify that randomly shuffling the text sets ensures a more diverse set of words, reducing the influence of irrelevant items that might be contained in the original set.

## 4.4 ABLATIONS

**Robustness against text memory** We investigate the robustness of our method against the text memory by excluding the target classnames contained in the set. As shown in Table 5, the performance is comparable, indicating that the key feature of our method is about aggregating the coarse-grained text semantics, which does not rely on abundant or large amount of text knowledge. Regarding that no explicit relation between an image and a classname is given in advance, the result also verifies that the model cannot find a shortcut to associate this information during training.

**Comparison on the number of retrieved words $k$.** In Figure 4, we examine the effect of the number of nearest neighbor words $k$ by differing the value with 5, 15, 20 and 30. The performance is comparable among $k = 5, 10, 15$, and $20$, while showing the best at $k = 20$. However, we notice that using redundant words during training causes confusion and ultimately leads to a performance drop.

**Qualitative results of the retrieved text set.** In Figure 5, we present a $k$-nearest neighbor retrieval results of our model. The retrieved words are ordered by its similarity score starting from the top-left. Highlights indicate the attention value computed by a cross-attention module. We only highlighted words with an attention value higher than 0.1.

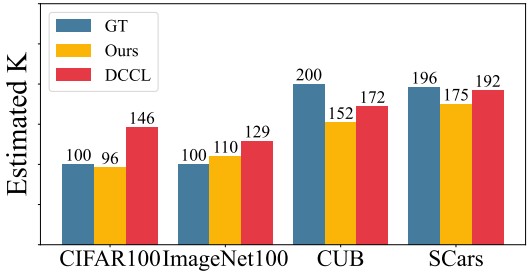 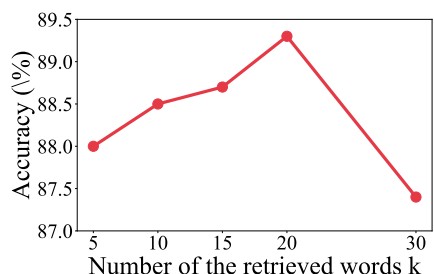

Figure 3: Results of the estimated number of classes $K$ on different benchmarks.

Figure 4: Results on different # of retrieved words

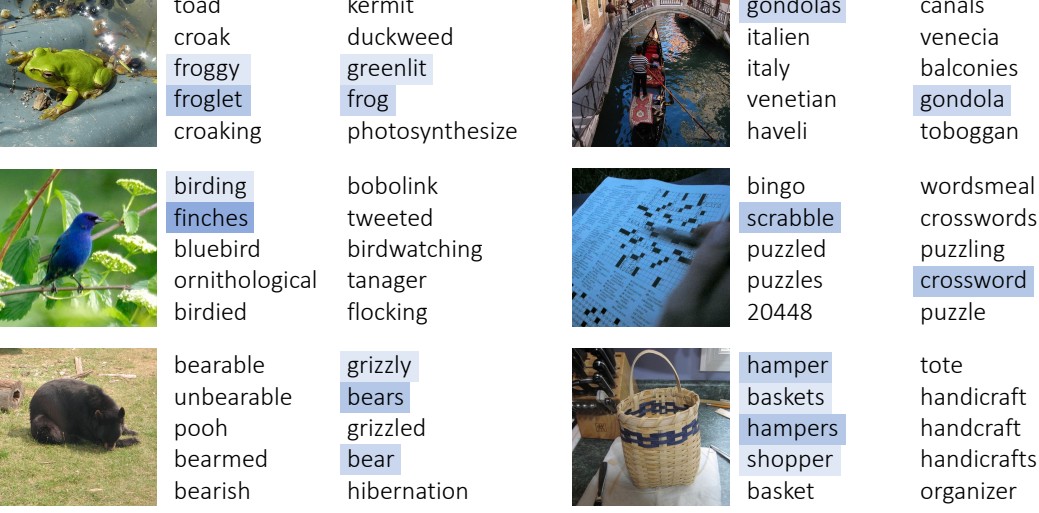

Figure 5: Qualitative results on $k$NN retrieved words.

## 5 CONCLUSION

We have presented a language-guided clustering method for GCD. Starting from the observation that the textual concepts exhibit a more distinguished clustering than the continuous visuals concepts distribution in real world, we introduce a multi-modal learning objective with linguistic grounding in the absense of the annotation visual grounding. To this end, the conventional $k$-nearest neighbor search is revisited to retrieve textual information from an arbitrary word corpus. The proposed learning loss encourages the retrieved text and the image paired to be close to each other, where the text effecitvely bridges the scattered images well clustered. In addition to the embedding learning, we propose to cluster without knowing the ground-truth number of clusters, which is fundamental assumption but often breaked in the previous work. In experiments, our method achieves state-of-the-art GCD performance on four public benchmarks among six of them.

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
