# OpenReview forum: "Visual Category Discovery via Linguistic Anchoring"
_ICLR.cc/2024/Conference — ICLR 2024 Conference Withdrawn Submission_

### Official Review · Reviewer_9cFD · 2023-10-31

**Soundness:** 3 good
**Presentation:** 3 good
**Contribution:** 2 fair
**Rating:** 3
**Confidence:** 5

**Summary:**

The paper introduces a simple but effective method for generalized category discovery (GCD). The authors make use of a pretrained vision-language model, i.e., CLIP, and propose to discover the novel classes by searching for the nearest words. They formalize a Vision-and-Language Contrastive Loss to achieve this goal. The experimental results on six widely-used GCD benchmarks demonstrated the superiority of the proposed method.

**Strengths:**

- The paper is clearly written and easy to follow.
- The proposed method is evaluated to be effective.

**Weaknesses:**

- The proposed method relies on an assumption that the classes (including known and unknown ones) can all be described using single English words, which I think might not always hold true, especially in the context of class discovery. As our goal is to discover unknown novel classes, chances are we encounter something completely new -- which are naturally hard to be verbalized, or even beyond our own human vocabulary. In this regard, the practicability of the proposed method may be limited.
- The proposed method is potentially similar to [a], yet without no proper discussion.
- Important baselines are missing, such as [b].

[a] CLIP-GCD: Simple Language Guided Generalized Category Discovery. 2023

[b] Parametric Classification for Generalized Category Discovery: A Baseline Study. ICCV 2023

**Questions:**

What is the impact of the used text corpus on the final performance? For instance, how does it influence the performance with respect to the topic (coverage), quantity, diversity, etc. of the text corpus in use?

---

### Official Review · Reviewer_ixQ1 · 2023-11-01

**Soundness:** 3 good
**Presentation:** 3 good
**Contribution:** 2 fair
**Rating:** 6
**Confidence:** 4

**Summary:**

The study introduces a novel approach to generalized category discovery (GCD), which classifies images from partially labeled datasets without prior knowledge of the total number of classes. This method leverages the interplay between visual and textual data through an image-text foundation model, like CLIP, to recognize new categories across both training and testing phases. Unlike traditional GCD methods that require pre-knowledge of category counts, this technique excels in determining the number of categories independently. It employs language-anchored contrastive learning, using a joint embedding space for images and text, and enforces image-text consistency constraints. The process is facilitated by a unique mechanism that assigns text embeddings to images via nearest-neighbor word retrieval from a diverse corpus and cross-attention aggregation, all without needing manual annotations. This language-anchored approach has demonstrated superior performance on established benchmarks.

**Strengths:**

1-The paper presents an innovative use of the CLIP model, utilizing it to aid in the discovery of new categories by retrieving relevant words, a technique that is unique to this study.

2-The experimental design is thorough and transparent, with the inclusion of comparative analysis against other methods using CLIP, ensuring that the evaluations are equitable.

3-The paper thoroughly explores and reports on ablation studies, providing a comprehensive understanding of the method's components and their individual contributions.

**Weaknesses:**

1-The method shows particular dependence on prior exposure of the images to the image-text foundation model, such as CLIP. This is evident as it fails to surpass the state-of-the-art (SoTA) for datasets like Aircraft, which presumably are less represented in CLIP's pre-training, whereas it excels with well-represented datasets such as ImageNet.

2- While the application of the proposed method to generalized category discovery (GCD) is novel, the overall approach is rather straightforward and lacks significant technical innovation. It mirrors techniques already prevalent in zero-shot and multimodal learning.

3-The issue of open vocabulary, which is closely related to the challenges addressed in the paper, is not discussed in the context of related work. This omission overlooks a pertinent area that should be considered for a comprehensive literature review.

4- The presentation of trends in Table 1 could be clearer.

Minor:
Isn't section 4.3 part of ablations?

**Questions:**

1- Regarding equation 1, could the model be adapted to incorporate supervised contrastive learning by including text prototypes in relation to class names?

2-In reference to equation 2, what would be the outcome if, instead of aggregating all positive examples for each modality, we evaluated them individually and then combined the losses? Could this approach address the issue where high textual similarity might permit visual embeddings to remain distant from one another?

3-Is it possible to weigh some words more favorably, for instance, the corresponding attributes for each image in the CUB or aircraft datasets?

---

### Official Review · Reviewer_4NxT · 2023-11-01

**Soundness:** 2 fair
**Presentation:** 3 good
**Contribution:** 2 fair
**Rating:** 3
**Confidence:** 3

**Summary:**

Generalized Category Discovery problem is studied in this paper, with the aim to discover new classes in a dataset and classify the corresponding instances. This paper finds building an intermediate text representation (referred to as "text anchor") is able to model semantic similarity between images better than visual features. Such text anchor feature is built by retrieval the k-nearest words in a large word corpus (Google Conceptual Cpations 12M dataset) in CLIP space. With the proposed text anchor feature, a contrastive learning framework is designed to preserve image-image, image-text consensus. Experimental results are promissing.

**Strengths:**

1. This paper is well written. It is easy to follow.
2. Resorting to CLIP to search for k most related word from a large corpus for each image is interesting idea.

**Weaknesses:**

1. The main concern of this paper is that the proposed methods and experiments do not fit into the conventional setup of Generalized Category Discovery (GCD), which assumes a finite labeled set. In this context, besides the limited known class labels, one should avoid introducing additional label information from external data or models. For instance, in the original paper by "Vaze et al., Generalized Category Discovery, CVPR 2022," the authors deliberately chose to initialize the image encoder with self-supervised model weights from ImageNet, rather than using the supervised counterpart. This approach ensures that knowledge from ImageNet labels does not leak into their proposed model or framework. However, this paper heavily relies on the CLIP model, and its knowledge, particularly a weak form of external labels, can be easily transferred to the proposed method. Therefore, the experimental setup in this paper differs from the original work in its design.

Furthermore, it is probable that the "text memory" employed in this paper already encompasses both the known and unknown class names in GCD. Consequently, this paper is more like performing image tagging first, which is highly likely to have already identified all unknown classes and achieved a precise classification. It harnesses the capabilities of CLIP without necessitating additional learning.

2. The use of different names to refer to the same term is causing some confusion. For example, "prototype" and "anchor" as well as "text memory" and "work memory."

3. Regarding the source for building the "text memory," it is repeatedly mentioned that any "arbitrary word corpus" will work. Unfortunately, this claim is unsupported, as no ablation study has been conducted. In fact, the "Google Conceptual Captions 12M dataset" is the only source used for this purpose.

4. Some related work is missing. When it comes to searching for k-nearest neighbor-related words to form the text prototype feature, it shares similarities with a line of early work that searches for k-nearest neighbors in the visual space and propagates associated tags to the query image. An example of this work is "Li et al., Learning social tag relevance by neighbor voting, TMM 2009."

**Questions:**

1. Could authors explain what is being memorized by "text memory"?  The reviewer's understanding is that it is fixed during the training/test.

2. Does the proposed frame generate for each image a pair of image (visual) prototype feature and a text prototype feature?  Or visual prototypes and text prototypes are a set of learned representatives w.r.t. the entire dataset with fixed number?

---

### Official Review · Reviewer_H3av · 2023-11-01

**Soundness:** 3 good
**Presentation:** 2 fair
**Contribution:** 3 good
**Rating:** 8
**Confidence:** 3

**Summary:**

This paper proposes a language-guided clustering method for generalized category discovery (GCD). The proposed approach pairs each image with a corresponding text embedding by retrieving k-nearest-neighbor words from a text corpus and aggregates them through cross-attention. The paper proposes also to perform clustering without knowing a priori the total number of classes.

The contributions of this paper are as follows:
- a language guided image clustering is proposed
- Clustering images into classes without knowing a priori the total number of classes.
- The proposed model achieves the state-of-the-art performance on the public GCD benchmark

**Strengths:**

- The proposed model achieves the state-of-the-art performance on the public GCD benchmark

- extensive evaluation of the model and the different technical decision

- Clustering images into classes without knowing a priori the total number of classes

**Weaknesses:**

The paper is hard to read and could use some rewriting.


Some part of the model are not very intuitive and not well described, for instance:

- How the linkage matrix is used for clustering images without requiring the number of classes?

- if the number of classes is not known, therefore some labels should be missing (otherwise, the number of labels should represent the number of classes). How the model is then able to classify images into these classes and associate a label with them?

**Questions:**

- How the linkage matrix is used for clustering images without requiring the number of classes?

- if the number of classes is not known, therefore some labels should be missing (otherwise, the number of labels should represent the number of classes). How the model is then able to classify images into these classes and associate a label with them?

---

### Author Response · Authors · 2023-11-21

Dear reviewers,

We appreciate your comments. While withdrawing this submission, we plan to submit an updated version of the paper to another conference. We will try to address your comments in the revision.
Thank you.